# Advances and Challenges in Antiviral Development for Respiratory Viruses

**DOI:** 10.3390/pathogens14010020

**Published:** 2024-12-31

**Authors:** Luis Adrián De Jesús-González, Moisés León-Juárez, Flor Itzel Lira-Hernández, Bruno Rivas-Santiago, Manuel Adrián Velázquez-Cervantes, Iridiana Monserrat Méndez-Delgado, Daniela Itzel Macías-Guerrero, Jonathan Hernández-Castillo, Ximena Hernández-Rodríguez, Daniela Nahomi Calderón-Sandate, Willy Salvador Mata-Martínez, José Manuel Reyes-Ruíz, Juan Fidel Osuna-Ramos, Ana Cristina García-Herrera

**Affiliations:** 1Unidad de Investigación Biomédica de Zacatecas, Instituto Mexicano del Seguro Social, Zacatecas 98000, Mexico; flor.lihe@gmail.com (F.I.L.-H.); rondo_vm@yahoo.com (B.R.-S.); iridiana941901@gmail.com (I.M.M.-D.); 36173990@uaz.edu.mx (D.I.M.-G.); 37183498@uaz.edu.mx (X.H.-R.); 20202095@uaz.edu.mx (D.N.C.-S.); willy_salvadormtz@hotmail.com (W.S.M.-M.); ana.garciaher@imss.gob.mx (A.C.G.-H.); 2Laboratorio de Virología Perinatal y Diseño Molecular de Antígenos y Biomarcadores, Departamento de Inmunobioquímica, Instituto Nacional de Perinatología, Ciudad de México 11000, Mexico; adrianvela18@gmail.com; 3Especialidad en Medicina Familiar, Unidad Académica de Medicina Humana y Ciencias de la Salud, Universidad Autónoma de Zacatecas, Zacatecas 98160, Mexico; 4Instituto Mexicano del Seguro Social, Unidad de Medicina Familiar # 4, Servicio de Medicina Familiar, Guadalupe, Zacatecas 98618, Mexico; 5Unidad Académica de Ciencias Químicas, Universidad Autónoma de Zacatecas, Zacatecas 98160, Mexico; 6Department of Infectomics and Molecular Pathogenesis, Center for Research and Advanced Studies (CINVESTAV-IPN), Mexico City 07360, Mexico; jonathan.hernandez@cinvestav.mx; 7División de Investigación en Salud, Unidad Médica de Alta Especialidad, Hospital de Especialidades No. 14, Centro Médico Nacional “Adolfo Ruiz Cortines”, Instituto Mexicano del Seguro Social (IMSS), Veracruz 91897, Mexico; jose.reyesr@imss.gob.mx; 8Facultad de Medicina, Región Veracruz, Universidad Veracruzana (UV), Veracruz 91700, Mexico; 9Facultad de Medicina, Universidad Autónoma de Sinaloa, Culiacán 80019, Mexico; osunajuanfidel.fm@uas.edu.mx

**Keywords:** antivirals, respiratory viruses, viral replication, entry inhibition, drug repurposing

## Abstract

The development of antivirals for respiratory viruses has advanced markedly in response to the growing threat of pathogens such as Influenzavirus (IAV), respiratory syncytial virus (RSV), and SARS-CoV-2. This article reviews the advances and challenges in this field, highlighting therapeutic strategies that target critical stages of the viral replication cycle, including inhibitors of viral entry, replication, and assembly. In addition, innovative approaches such as inhibiting host cellular proteins to reduce viral resistance and repurposing existing drugs are explored, using advanced bioinformatics tools that optimize the identification of antiviral candidates. The analysis also covers emerging technologies such as nanomedicine and CRISPR gene editing, which promise to improve the stability and efficacy of treatments. While current antivirals offer valuable options, they face challenges such as viral evolution and the need for accessible treatments for vulnerable populations. This article underscores the importance of continued innovation in biotechnology to overcome these limitations and provide safe and effective treatments. Combining traditional and advanced approaches in developing antivirals is essential in order to address respiratory viral diseases that affect global health.

## 1. Introduction

In recent decades, the development of antivirals to combat viral respiratory infections has advanced significantly, driven by the need to confront high-incidence and rapidly evolving pathogens, such as influenza, respiratory syncytial virus (RSV), and SARS-CoV-2 [1,2,3]. These viruses represent a considerable threat to global public health, with impacts ranging from mild infections to severe disease, especially in vulnerable populations such as children and older adults [4,5,6,7]. Antivirals play an essential role in controlling these infections by interrupting the viral replication cycle at crucial points, reducing viral load, and mitigating disease progression [1,8,9,10,11].

This paper comprehensively reviews advances in antiviral development, focusing on current treatments’ specific mechanisms of action and technological innovations redefining the field. Traditional approaches to inhibiting viral proteins and introducing therapies targeting host cellular components, as well as strategies that seek to decrease the likelihood of viral resistance, are discussed. In addition, we explore innovative techniques such as drug repurposing and advanced bioinformatics, allowing effective therapeutic compounds to be identified more quickly and at a lower cost. The review highlights the importance of employing molecular modeling and other biotech tools in drug design, aiming to optimize the affinity and selectivity of antivirals.

Finally, we discuss the challenges and limitations in developing antivirals, such as drug resistance, the rapid mutation of certain viruses, and the need for accessible and effective therapies for the most affected populations. In this analysis, we underline the importance of innovating antivirals and integrating multidisciplinary approaches that accelerate the development of safer and more effective treatments adapted to contemporary epidemiological challenges.

## 2. Antivirals: A Brief Overview

Antiviral therapeutics have experienced notable progress since the introduction of idoxuridine, the first antiviral used to treat Herpes simplex ocular infections. This drug, whose chemical structure mimics thymidine, interferes with viral replication by being incorporated in place of thymidine during the synthesis of viral DNA [12]. Despite having more than 90 antivirals available in the last four years, more than 200 viruses without specific antiviral treatments continue to present a global challenge, affecting populations worldwide [13].

Viral infections remain a significant threat to human health, further complicated by the shortage of approved vaccines and antiviral treatments [14]. Antivirals’ central strategy is to prevent the intracellular replication of the virus, acting in different phases of the replicative cycle, including the binding of viral proteins, their maturation, and the exocytosis of the virion (Figure 1) [15].

Antiviral treatments act in the different phases of the replicative cycle by inhibiting the main functional proteins of each phase, such as proteins that mediate binding to the host and cellular receptors, making it difficult for the virus to fusion and enter cells. Those focus on viral non-structural proteins, which interfere with viral replication, a critical step in the transcription and translation of viral genetic material [16]. Additionally, host-directed antivirals seek to inhibit cellular components essential to the viral cycle, reducing the likelihood of viruses developing drug resistance. These drugs can inhibit specific enzymes, prevent the entry of the virus into the cell and the assembly of the infection, and enhance a more effective immune response [17,18].

The therapeutic strategy of direct-acting antivirals focuses on specific viral targets to minimize unwanted interactions with host proteins. These antivirals have the potential to be applied in the treatment of various viral diseases, taking advantage of the similarity between specific essential viral proteins. Notable examples include daclatasvir, elbasvir, ledipasvir, and pibrentasvir, all of which have demonstrated efficacy in the context of particular diseases, suggesting a broad horizon for their potential repurposing in other viral infections [19,20].

The virus’s binding phase and entry into the host cell represent a critical target. The inhibition of this process prevents the initial establishment of the infection for viruses such as HCV (Hepatitis C virus), Dengue virus (DENV), and Immunodeficiency virus (HIV). Drugs such as umifenovir target this phase by interfering with this binding mechanism [21].

On the other hand, viral protease inhibitors constitute another critical class of antivirals, specifically directed at viruses with a protein envelope. They block the protease enzymes necessary for the processing of viral polyproteins, preventing the assembly of the virus and the generation of the different viral proteins (structural and non-structural). Furthermore, these drugs can also prevent the generation of viral particles, preventing the formation of structural proteins and virion assembly and inhibiting viral replication [22,23].

Finally, inhibitors of the replication of genetic material play a crucial role by interfering with the enzymes involved in viral DNA or RNA replication. This approach seeks to prevent the multiplication of the viral genome within the host cell, addressing both DNA and RNA viruses and highlighting the diversity in antiviral development strategies [24,25] (Figure 2).

## 3. Overview of Human Respiratory Viruses

Human respiratory viruses constitute a diverse group of pathogens that predominantly affect the respiratory system, from the nose and throat (upper respiratory tract) to the lungs (lower respiratory tract). These infectious agents are responsible for various illnesses, ranging in severity from a simple common cold to more serious diseases such as pneumonia, bronchiolitis, and severe acute respiratory syndrome. The transmission of these viruses generally occurs through respiratory droplets expelled when coughing, sneezing, or talking, although some can be transmitted through direct contact with contaminated surfaces (Table 1).

Respiratory viruses, including influenza, respiratory syncytial virus (RSV), SARS-CoV-2, rhinoviruses, and adenoviruses, significantly impact global health. This review focuses on RSV, influenza, and SARS-CoV-2, the only respiratory viruses with approved antiviral therapies for clinical use. In contrast, other respiratory viruses lack specific antiviral treatments, relying solely on supportive care or investigational therapies, underscoring a critical gap in antiviral development.

This review analyzes antiviral drugs, their mechanisms, and clinical applications by narrowing its scope to viruses with approved treatments. It also offers a structured and focused discussion of current strategies for combating respiratory diseases.

## 4. Antivirals for Influenza Viruses

The influenza virus exhibits significant diversity, with both IAV and B (IBV) viruses circulating in recent years. IAV, in particular, is classified into several subtypes based on the variability of its outer membrane proteins: hemagglutinin (HA), which has 18 types, and neuraminidase (NA), which has 11 types; and the matrix 2 ion channel protein (M2) only shows 1 type [37]. The M2 and NA proteins have been the research focus for general antivirals; recently, an antiviral targeting the influenza polymerase complex (IVC) has been developed [38]. While M2 antivirals, known as adamantanes (amantadine and rimantadine), function by blocking M2 ion channels to prevent the flow of H^+^ into the virion, their use was recommended by the FDA until 2019. However, due to the emergence of resistance, the focus has shifted to using neuraminidase inhibitor antivirals (NAIs), also known as viral release inhibitors, as well as antivirals that target IVC replication [38,39].

### 4.1. Viral Release Inhibitor

For IAV, NA plays a crucial role in regulating the release of virions alongside N-acetyl-neuraminic acid. Due to their structural similarity to NA substrates, NAIs are effective against IAV and IBV [40]. Among the NAIs approved by the FDA are oseltamivir, zanamivir, and peramivir [39,41] (Figure 2).

#### 4.1.1. Zanamivir

Due to the resistance associated with adamantane antivirals for IAV and their low efficacy against IBV, new antivirals targeting the NA protein were developed and have been effective against IBV since 1993 [39,42,43]. The first NA inhibitor approved by the FDA was zanamivir, which was introduced in 1999. Its structure is based on 2-deoxy-2,3-didehydro-N-acetylneuraminic acid, commonly referred to as DANA. Zanamivir can be administered intranasally, intraperitoneally, or intravenously [41,44]. Studies have shown that this medication can prevent the onset of infection and reduce the duration of symptoms [45]. The primary medical indication for zanamivir is its inhalation use in patients with influenza, but it can be administrated intravenously as well. The FDA recommends it for adults and pediatric patients (≥7 years in Canada and the United States, and ≥5 in most countries, including the European Union and Australia) [44,46].

#### 4.1.2. Oseltamivir

This drug is designed based on zanamivir, incorporating its DANA structure while adding a cyclohexene ring, a C4 amino group, and a lipid side chain. These modifications enable oseltamivir to be administered orally, unlike zanamivir, which is delivered via inhalation and poses challenges for patients with respiratory issues [44,47].

Oseltamivir is available in capsule and liquid suspension forms, allowing at least 80% of an oral dose to reach systemic circulation as the active metabolite. Furthermore, both the EMA and FDA have approved its use for individuals ranging from neonates to adults [44,46].

#### 4.1.3. Peramivir

This DANA-derived antiviral features a cyclopentane ring, a guanidino group, and a hydrophobic side chain, enabling it to target IAV variants resistant to zanamivir and oseltamivir. It is administered intravenously [41,44]. Notably, this antiviral is the only NAIs approved by the FDA for treating acute uncomplicated influenza [48]. During the 2009–2010 pandemic, the FDA authorized the emergency use of peramivir [49]. Substantial in vitro evidence indicates that peramivir exhibits greater efficacy against the NA protein in IAV and IBV infections than zanamivir and oseltamivir [50]. Peramivir is a single-dose treatment, and the FDA recommends its use for adults and pediatric patients (≥6 years). However, clinical studies have demonstrated its efficacy in pediatric patients as young as 2 years [46,51].

### 4.2. Replication Inhibitor

Antivirals targeting the RNA-dependent RNA polymerase (RdRp) in IAV are extensively studied due to the enzyme’s crucial role in the replication and transcription of RNA viruses and its high conservation across IAV variants. The FDA has approved using baloxavir carboxyl for this purpose [38,46] (Figure 2).

#### Baloxavir Marboxil

This antiviral targets the endonuclease activity of the PA subunit, preventing viral mRNA transcription [52]. Since its approval by the FDA in 2018, baloxavir has been indicated for use in adults and pediatric patients (≥12 years) [46]. Several clinical trials have demonstrated its effectiveness against IAV and IBV, showing significant efficacy with a single dose. Notably, it has been shown to reduce fever and the viral load within just 24 h, outperforming NAIs in various clinical assessments [53,54].

Although the FDA has approved these antivirals, new drugs are still being researched and evaluated in the clinical phase for treating IAV and IBV, as these viruses have demonstrated resistance and genetic variability, especially IAV.

## 5. Antivirals for Respiratory Syncytial Virus (RSV)

RSV primarily affects specific segments of the population, including pediatric patients (<5 years) and older adults (>65 years). Clinical trials of different antivirals have shown a lack of antiviral efficacy in real-world settings, but there is currently only one FDA-approved antiviral for RSV: ribavirin. Several antiviral studies have been canceled for various reasons, including strategic decisions, low participation, and safety concerns. Multiple antiviral compounds are currently being developed at different clinical stages [55,56] (Figure 2).

Ribavirin works by increasing the mutation rate of the viral polymerase during RSV replication, which directly impacts the production of the virus [57]. As the only antiviral approved by the FDA for RSV, it has been shown to significantly reduce symptoms in pediatric patients during the early stages of infection. However, its use is limited in this population due to potential side effects, including bronchospasm, dyspnea, teratogenic effects, and the high cost of treatment [58,59].

However, various strategies have been implemented to combat RSV infection, including antibodies explicitly targeting the virus. The FDA approved the monoclonal antibody nirsevimab in 2023 to prevent lower respiratory tract infections in pediatric patients (<2 years) and high-risk individuals. It is recommended for newborns during their first winter season when RSV is circulating. A vital advantage of this approach is that the antibodies are directed at a viral protein; in this case, nirsevimab targets RSV protein F, which helps reduce the virus’s entry into cells [59,60,61].

## 6. Antivirals for SARS-CoV-2

The entry of SARS-CoV-2 involves binding and inserting into the host cell, where it produces viral proteins to initiate replication. During these processes, there are specific points where antivirals can act. Various antivirals were evaluated after the COVID-19 health emergency to treat SARS-CoV-2 infection. These drugs have been classified according to their point of action into antivirals that are polymerase replication inhibitors; antivirals that are protease inhibitors; and antivirals that are virus entry inhibitors [62].

### 6.1. Replication Inhibitors

SARS-CoV-2 replication is driven by RNA-dependent RNA polymerase (RdRp). RdRp inhibitors or drugs include remdesivir, molnupiravir, favipiravir, and ribavirin [62] (Figure 2).

#### 6.1.1. Remdesivir

During the COVID-19 pandemic, remdesivir was declared the most efficient drug for inhibiting the viral RNA-dependent RNA polymerase (RdRp) of SARS-CoV-2 [63]. In addition, it was the first antiviral to be approved by the FDA to treat COVID-19. Later, in 2020, it was approved for use in adults and pediatric patients (≥12 years) with mild to moderate cases [64,65]. Later, in 2022, it was approved by the FDA for treating pediatric patients (≥28 days) [66].

Remdesivir is a prodrug; its conversion to the active metabolite GS-443902 triphosphate confers its antiviral activity by being able to incorporate it into the viral RNA, leading to premature termination [66]. A clinical study showed that remdesivir is one of the best drugs to shorten the recovery time in adults hospitalized with COVID-19 [64]. However, this compound could face some challenges since there are data on the resistance to remdesivir in an in vitro virus culture, suggesting that it occurs due to multiple amino acid substitutions in the viral polymerase. Although resistance levels to remdesivir have been very low so far, they can be overcome by using higher, non-toxic concentrations of remdesivir [67].

#### 6.1.2. Molnupiravir

Molnupiravir, also known as EIDD-2801/MK 4482, is a prodrug that is converted into its active form, EIDD-1931, once inside the cell. This metabolite is incorporated into the viral RNA during replication, leading to viral genome errors that inhibit replication [64].

Initially developed for the Venezuelan equine encephalitis virus (VEEV), molnupiravir was later found to have broad-spectrum antiviral activity against respiratory viruses, including influenza and SARS-CoV-2. During the COVID-19 pandemic, the US FDA granted it Emergency Use Authorization (EUA) in December 2021 for treating COVID-19. However, its use is contraindicated during pregnancy due to potential teratogenic effects [68].

Molnupiravir induces mutagenesis by introducing base-pairing errors in the viral RNA, a mechanism shared with ribavirin. This mutagenesis disrupts viral replication by generating defective progeny virions. The similarity in their mechanisms of action has prompted ongoing studies to evaluate their efficacy and potential risks in SARS-CoV-2 infections [69].

#### 6.1.3. Favipiravir

Favipiravir is a prodrug converted intracellularly to favipiravir-ribofuranosyl-5′-triphosphate, inhibiting viral RNA polymerase [69]. Favipiravir has antiviral effects against several RNA viruses. It has shown the advantage of significantly reducing patients’ recovery time with mild to moderate COVID-19 [70]. In addition, previous studies have shown that it reduces the viral load of SARS-CoV-2 in the upper respiratory tract and lungs. Moreover, it is currently in clinical trials for treating COVID-19, awaiting approval in the US; however, it has already been approved in other countries such as Russia, Japan, and India [69].

#### 6.1.4. Ribavirin

Ribavirin is a competitive inhibitor of the enzyme inosine monophosphate dehydrogenase. It prevents the production of guanosine monophosphate (GMP) and thus inhibits viral RNA and DNA replication [62].

Like molnupiravir, ribavirin induces mutagenesis in viral RNA. While this mechanism has demonstrated efficacy against several viruses, ribavirin has not yet received FDA approval for treating COVID-19. Clinical trials are ongoing to assess its role in SARS-CoV-2 infections [69].

### 6.2. Protease Inhibitors

SARS-CoV-2 has two types of viral proteases: 3-chymotrypsin-like cysteine protease (3CLpro or Mpro, main protease) and papain-like serine protease (PLpro). Various antivirals seek to inhibit these proteases [62] (Figure 2).

#### 6.2.1. Paxlovid

Paxlovid is composed of two different drugs, nirmatrelvir and ritonavir. Ritonavir was used previously in HCV and HIV infections. It was developed by Pfizer and consists of two oral tablets corresponding to Nirmatrelvir, which acts directly on the NSP5 protease (3C-like; 3CL) of SARS-Cov-2, and Ritonavir, which is described as a pharmacological enhancer since it prevents the mechanism of the cytochrome P450 3A4 enzyme, responsible for processing Nirmatrelvir, which leads to the greater bioavailability of the active molecule and, therefore, a more significant effect [71].

Nirmatrelvir binds to the catalytic site of the NSP5 protease, specifically to the Cys145 residue, inhibiting the processing of polyproteins and preventing viral replication. Paxlovid is eliminated via the kidneys, which has been linked to some renal intoxication events [72].

Paxlovid has demonstrated a 56% reduction in the risk of hospitalization or death from COVID-19 (76%) when administered within 5 days of symptom onset, compared to a placebo. Paxlovid is approved for treating mild to moderate cases of COVID-19 in people aged 12 years or older, weighing at least 40 kg, who have tested positive for SARS-CoV-2. This approval is intended for those individuals who face an elevated risk of progressing to severe COVID-19, which could lead to hospitalization or even a fatal outcome. The dosing regimen consists of 300 mg of nirmatrelvir with 100 mg of ritonavir, administered twice daily for 5 days. Adverse events reported are dysgeusia (4.8%), diarrhea (3.9%), and vomiting (1.3%) [71,73,74].

#### 6.2.2. Ensitrelvir

Ensitrelvir (Xocova) is an oral antiviral developed by Shionogi & Co., Ltd. (Osaka, Japan) in collaboration with Hokkaido University. Its mechanism of action is based on the selective inhibition of the SARS-CoV-2 3CL protease. By blocking this protease, ensitrelvir prevents the maturation of viral proteins necessary for producing new infectious particles, thus stopping the spread of the virus [75].

In March 2022, Xocova received emergency regulatory approval in Japan to treat COVID-19. Subsequently, in March 2024, it obtained standard approval in the same country, supported by data demonstrating its clinical efficacy in reducing typical COVID-19 symptoms and its antiviral capacity in patients with mild to moderate SARS-CoV-2 infections [76].

Regarding its status in the United States, ensitrelvir received Fast Track designation from the Food and Drug Administration (FDA) in April 2023 to expedite its development and review due to unmet medical needs in treating COVID-19. However, it has not received full approval or emergency use authorization from the FDA and remains an investigational drug outside Japan [77,78].

Clinical trials have indicated that ensitrelvir effectively reduces viral load and improves symptoms associated with COVID-19, especially in patients with mild to moderate infections. A post-marketing study in Japan, which included more than 4000 patients, evaluated the safety and effectiveness of ensitrelvir in clinical practice, providing additional data on its real-world safety and efficacy profile [79,80,81].

It is important to note that, although the preliminary results are promising, the clinical efficacy of ensitrelvir is still under evaluation in various countries. The drug’s approval and availability will depend on the results of additional studies and the decisions of the relevant health authorities [82,83].

#### 6.2.3. Nelfinavir

Also known as Viracept, nelfinavir is an antiviral medication used to treat HIV infections. It inhibits retroviral proteases essential for replicating and releasing mature viral particles. Nelfinavir has been found to inhibit the activity of the main protease of SARS-CoV-2. However, it is also a potent inhibitor of cell fusion caused by the S glycoprotein [84,85,86].

In vitro studies have shown that nelfinavir inhibits SARS-CoV-2 replication, making it a promising drug for treating coronavirus infection. While studies continue, approval remains pending [69].

#### 6.2.4. Ritonavir and Lopinavir

Ritonavir and lopinavir are protease inhibitors initially developed to treat HIV by targeting the active site of the viral protease. Ritonavir is often combined with lopinavir to enhance its pharmacokinetics and pharmacodynamics [63,69]. While early studies suggested potential antiviral activity against SARS-CoV-2, more recent data indicate that their clinical benefits, if any, are not directly related to antiviral effects. In vitro assays have shown limited potency against SARS-CoV-2, and large-scale clinical trials did not demonstrate significant efficacy in reducing disease severity or improving outcomes in COVID-19 patients [87,88,89,90,91].

Despite these limitations, ritonavir gained renewed attention as part of paxlovid, a combination therapy with nirmatrelvir. This combination leverages ritonavir’s role as a pharmacokinetic enhancer rather than a direct antiviral agent [72,92,93].

#### 6.2.5. Atazanavir and Darunavir

Atazanavir (Reyataz) and darunavir (Prezista) are HIV protease inhibitors with mechanisms of action like lopinavir and ritonavir. Both drugs have been investigated in vitro and clinical trials for potential activity against SARS-CoV-2, either alone or in combination with other agents such as nitazoxanide, ritonavir, dexamethasone, or daclatasvir [69,94].

However, the evidence for their efficacy in treating COVID-19 remains inconclusive. Neither atazanavir nor darunavir has been approved for this purpose, and their antiviral effects against SARS-CoV-2 have yet to be verified in clinical settings. As such, further research is needed to determine their therapeutic potential and safety profiles for COVID-19 [95,96,97].

### 6.3. Inhibitors of Virus Entry

Inhibitors of virus entry are designed to block the initial steps of viral infection by targeting the virus’s binding and fusion mechanisms with host cell receptors. These inhibitors prevent the virus from attaching to the host cell surface or fusing with the cell membrane, thus blocking the entry process entirely. This strategy is pivotal because preventing viral entry can halt the infection before it begins its replication cycle, significantly reducing the viral load and limiting disease progression [62] (Figure 2).

#### 6.3.1. Hydroxychloroquine

At least 80 trials have been reported worldwide with chloroquine, hydroxychloroquine, or both, sometimes in combination with other drugs [70]. Hydroxychloroquine is an immunosuppressive agent used for various autoimmune disorders and as a potent antiparasitic drug. It works by inhibiting terminal glycosylation of the ACE2 receptor, which inhibits virus entry, infection, and progression [98]. Early studies suggested that hydroxychloroquine might reduce the risk of thrombosis, a significant complication in COVID-19, and prevent the exacerbation of pneumonia, potentially shortening the course of the disease [63,69].

Based on these preliminary findings, the FDA granted an Emergency Use Authorization (EUA) for hydroxychloroquine on 28 March 2020, to treat COVID-19. However, subsequent large-scale clinical trials and meta-analyses demonstrated that hydroxychloroquine did not significantly improve clinical outcomes, reduce viral load, or prevent severe disease progression. These findings, coupled with documented adverse effects such as anxiety, insomnia, gastrointestinal disturbances, and cardiomyopathies, led to the FDA revocation of the EUA in June 2020 [63,69].

The WHO reviewed the clinical evidence and concluded that hydroxychloroquine should not be recommended for treating COVID-19. This decision was based on the lack of efficacy observed in randomized controlled trials and the risks associated with its use, particularly cardiovascular toxicity [99]. Hydroxychloroquine’s inability to effectively modulate the overactivation of the innate immune system further limited its clinical utility [63].

#### 6.3.2. Arbidol

Also known as umifenovir, Arbidol is an indole-based antiviral drug approved for treating influenza virus in China and Russia. However, the FDA does not currently approve it in the U.S. Regarding its mode of action, Arbidol decreases the interaction between viruses and the host during endocytosis and exocytosis, interrupting multiple phases of viral cycle replication. It is currently in clinical trials for various diseases, including SARS-CoV-2 [69].

#### 6.3.3. APN01

APN01 is a human recombinant ACE2 developed for treating pulmonary arterial hypertension, acute lung injury, and acute respiratory distress syndrome. It is known that SARS-CoV-2 enters human cells through the ACE2 receptor, so APN01 acts by preventing the interaction of this receptor with the virus, thus minimizing lung injury and multiple organ dysfunction. APN01 is currently in clinical trials in several countries to treat SARS-CoV-2 [69].

Although many of these antivirals continue to be analyzed and complete all the clinical studies necessary for their approval, they are not the only ones under evaluation. Various new and retargeted antivirals are being investigated for the treatment of COVID-19. The scientific community continues to explore different approaches and potential therapies to combat the virus in hopes of finding more effective and accessible options for patients affected by the disease.

## 7. Clinical Resistance to Antivirals: Mechanisms and Solutions

Antiviral resistance is a significant challenge in treating viral infections. It arises when viruses adapt to evade drug action, reducing efficacy and limiting therapeutic options. RNA viruses like influenza and RSV are particularly prone to resistance due to their high mutation rates and the lack of proofreading mechanisms in their replication machinery. SARS-CoV-2, while exhibiting high mutation rates, possesses an exoribonuclease proofreading function in its Nsp14 protein, which helps maintain genome fidelity [100].

Resistance primarily results from genetic mutations that alter viral targets, reducing drug affinity. For example, mutations in the M2 protein of influenza confer resistance to amantadine, while changes in the neuraminidase gene reduce susceptibility to oseltamivir. Similarly, mutations in the RNA-dependent RNA polymerase (RdRp) of SARS-CoV-2 can decrease the efficacy of remdesivir. Viruses may also evade entry inhibitors by modifying surface proteins, such as the hemagglutinin in influenza and the spike protein in SARS-CoV-2, reducing the effectiveness of monoclonal antibodies [101,102,103,104].

Resistance leads to treatment failure, increased disease severity, and the spread of resistant strains, complicating disease control efforts and rising healthcare costs. For instance, oseltamivir-resistant influenza strains and monoclonal antibody-resistant SARS-CoV-2 variants have emerged, highlighting the need for ongoing vigilance [105,106,107].

Key approaches include combination therapies targeting multiple viral mechanisms, prudent use of antivirals to minimize selective pressure, and genomic surveillance to detect emerging resistance mutations. Developing next-generation antivirals, such as broad-spectrum agents or therapies targeting host factors, offers promising solutions [8,108,109,110,111,112].

Addressing antiviral resistance requires a multifaceted approach combining innovation, surveillance, and responsible drug use. These strategies are essential in order to preserve antiviral efficacy and improve clinical outcomes in combating viral infections.

## 8. Novel Approaches to the Discovery of Antivirals

The continued evolution of viruses and the emergence of new strains pose significant challenges to public health. The rapid spread of viral diseases, such as influenza, HIV, Ebola, and, more recently, SARS-CoV-2, highlights the urgent need to develop effective treatments. However, the search for drugs represents a significant challenge in terms of time and resources. In this sense, drug repurposing has emerged as an attractive alternative, taking advantage of the availability of molecular data and advanced bioinformatics tools.

### 8.1. Drug Repositioning

Drug repositioning involves the systematic screening of approved or investigational compounds to determine their efficacy in diseases other than those for which they were initially developed [113]. This approach relies on the ability of many drugs to affect multiple biological pathways, making them potentially useful in various medical conditions.

The drug repositioning process involves several stages. First, an unmet medical need is identified. Then, the pharmacological and molecular targets on which the drug will act to produce the desired therapeutic effect are identified [114].

The validation and evaluation of the drug involves checking whether the selected target is related to the disease sought to be treated. Subsequently, we proceed with identifying compounds through biological assays, such as high-throughput screening (HTS), which allows the evaluation of libraries of compounds against a biochemical assay in cells. Another approach includes small molecule fragment screening, where libraries of fragments are screened to identify compounds with activity. Finally, lead validation is carried out, where analogous compounds are evaluated to establish structure–activity relationships, target selectivity, and the selected drug’s physicochemical and pharmacokinetic properties (ADME: Absorption, Distribution, Metabolism, and Excretion). [115].

#### 8.1.1. Advantages of Drug Repositioning

Drug repositioning presents significant advantages compared to the discovery of new compounds. These include reduced development costs and times and an increased likelihood of clinical success due to prior safety and pharmacokinetic data availability. By rescuing drugs already approved by the FDA or other regulatory entities, much of the uncertainty associated with developing new drugs is avoided, thus accelerating the process of bringing therapies to market [116].

#### 8.1.2. Bioinformatics in Drug Repositioning

Bioinformatics plays an essential role in all stages of the drug repositioning process. Integrating biological and chemical data with advanced analysis techniques allows identifying potential interactions between drugs and molecular targets, facilitating understanding disease mechanisms and supporting the development of targeted therapies for specific viral diseases [117]. Some bioinformatics tools commonly used in this context include protein structure analysis, the study of biological networks, protein–protein interactions, genomic data mining, and biomedical ontologies and knowledge graphs, which help understand complex biological systems and facilitate drug discovery [117,118].

In the drug repositioning process, various bioinformatics tools and software analyze molecular data and make accurate predictions. For example, AutoDock Vina is widely used for molecular docking, allowing researchers to accurately model the interaction between a drug and its molecular target. However, it is essential to note that many other kinds of docking software are available, such as GOLD, Glide, and FlexX, which offer alternative methods and may vary in performance depending on the specific research needs [119,120,121,122].

Benchmarking studies, such as those comparing docking performance (e.g., Chachulski, L. (2024) [123]. Development and Application of Compound Class-Specific Benchmark Data Sets for Differentiated Assessment of Docking and Scoring Algorithm Performance. PhD Thesis, Constructor University), provides valuable insights into the strengths and limitations of different tools. Furthermore, molecular modeling tools such as Open Babel and PyMOL facilitate the visualization and manipulation of molecular structures, which can support docking exercises. However, they are not directly involved in the docking task itself. For a successful docking study, following best practices, such as carefully preparing the ligand and receptor structures, choosing the appropriate docking algorithm, and critically analyzing the results, is essential [123,124,125]. Key studies in the field, such as those by Morris et al. (2009), highlight these best practices and guide for performing accurate and reliable docking experiments. In their work, ‘AutoDock4 and AutoDockTools4: Automated docking with selective receptor flexibility’, Morris and colleagues describe the importance of proper receptor preparation, selecting suitable docking parameters, and validating the docking results for reliable predictions [126].

Databases such as DrugBank and PubChem provide detailed information on chemical compounds and their properties. In contrast, protein banks such as the Protein Data Bank (PDB) contain three-dimensional structures of proteins relevant for drug development. These three-dimensional structures are obtained using techniques such as cryo-electron microscopy (cryoEM), X-ray crystallography, and nuclear magnetic resonance (NMR). These combined tools and resources allow researchers to effectively explore and evaluate a broad spectrum of compounds and molecular targets in their search for effective antiviral therapies [127,128,129].

For more information, we recently published a chapter on drug repurposing against emerging viruses, describing the importance of using molecular docking in antiviral discovery [130].

#### 8.1.3. The Role of Molecular Docking in Drug Repositioning

Molecular docking, a molecular engineering technique, is essential in drug repositioning. This methodology allows for predicting the three-dimensional structure of the binding between a drug and its molecular target, providing crucial information on the affinity and stability of the interaction. By modeling these interactions at the molecular level, molecular docking facilitates the selection of promising candidates for subsequent experimental testing, thereby optimizing the drug discovery process [113,116]. The binding free energy in molecular docking should be negative, indicating a favorable interaction between the ligand (drug) and the receptor (target protein).

The binding free energy in molecular docking should be negative, indicating a favorable interaction between the ligand (drug) and the receptor (target protein). The combination of drug repositioning and molecular docking significantly impacts the development of antivirals. By leveraging existing compounds and predicting precise molecular interactions, this strategy accelerates the identification of effective therapeutic candidates against viral diseases. Additionally, by optimizing drug efficacy and selectivity, molecular docking can help minimize side effects and viral resistance, thereby improving the clinical effectiveness of antiviral treatments and providing a preliminary approach to in vivo and ex vivo testing, and, later, in the clinical phase [131,132,133,134].

## 9. Innovations and Technological Advances in the Development of Antivirals

Designing new therapies for respiratory viruses involves addressing significant challenges, such as rapid viral evolution, airborne transmission, and increasing resistance to medications due to the genetic flexibility of RNA viruses. These characteristics complicate the development of long-lasting treatments, as RNA viruses readily mutate to evade host defenses and therapeutic agents [135]. Recent innovations focus on leveraging monoclonal antibodies, recombinant proteins, and nucleic-acid-based therapies to combat these challenges. These approaches prioritize enhancing efficacy, safety, stability, and bioavailability while ensuring affordability, particularly for vulnerable populations and developing regions disproportionately affected by respiratory viral diseases [136,137,138].

Nanotechnology has emerged as a vital tool for delivering and stabilizing active ingredients in antiviral treatments. This technology allows the precise targeting of the respiratory system, ensuring better drug stability and effectiveness, for instance, with nanoparticles and active compounds delivered intranasally or via nebulization bypass first-pass metabolism, enhancing therapeutic outcomes. These delivery methods also stimulate a localized immune response through lung-resident macrophages, which is crucial for detecting and neutralizing viral antigens [139]. Such innovations pave the way for treatments that are both efficient and accessible.

Recent advances have also highlighted the potential of inhalable vaccines for respiratory viruses. Wu et al. (2021) [140] developed an inhaled adenovirus-based vaccine for COVID-19, demonstrating acceptable safety, robust immunogenicity, and reduced adverse effects compared to intramuscular administration. This approach underscores the importance of exploring alternative administration routes to improve patient outcomes and reduce side effects. Additionally, nanobodies—antibodies composed solely of heavy chains—are being investigated for enhanced stability and effectiveness. Schoof et al. (2020) showed that these nanobodies bind effectively to the coronavirus spike protein, blocking viral entry through the ACE2 receptor. These findings represent promising steps toward more stable and efficient therapeutic options [141].

Research into stable formulations and excipients continues to advance antiviral therapies. Suberi et al. (2022) [142] demonstrated the use of non-inflammatory polymer-based formulations to deliver mRNA vaccines effectively to the lungs. These efforts aim to address the challenges associated with maintaining the stability and bioavailability of active ingredients, which are critical for ensuring the success of lung-targeted therapies [143].

Gene-editing technologies, particularly the CRISPR/Cas system, offer revolutionary potential in antiviral therapy. Initially identified as part of a bacterial adaptive immune system, CRISPR/Cas was repurposed to target viral genomes [143]. For example, CRISPR/Cas13 has been employed to inhibit influenza genomic material, while transgenic chickens have been engineered to express CRISPR RNA targeting avian influenza H5N1. In SARS-CoV-2, studies by Ryu et al. (2020) [144] and Abbott et al. (2020) [145] utilized RNA guides to target genes encoding the spike protein and non-structural proteins, effectively cleaving the viral genome to prevent replication. Despite its potential, the clinical application of CRISPR-based therapies requires further optimization and extensive trials to address safety and efficacy concerns [146].

Existing vaccines provide effective prevention for other major pandemic-causing viruses, such as influenza A virus (IAV) and respiratory syncytial virus (RSV). For example, nirsevimab, a monoclonal antibody, has shown efficacy against RSV [147], while multivalent vaccines target multiple influenza strains [142]. However, the high mutation rates of RNA viruses necessitate continuous research to develop novel treatments or repurpose existing therapies. This proactive approach is essential in order to stay ahead of health emergencies and address the persistent threat posed by zoonotic viruses that can overcome species barriers and develop resistance to current treatments.

## 10. The Future of Antiviral Treatments

Various antiviral treatments currently under evaluation include nucleoside analogues (NAs), natural compounds, and peptides. These alternatives need further exploration to address the challenge of viral resistance.

### 10.1. Nucleoside Analogues

NAs play a crucial role in treating diseases such as cancer and viral infections. They possess diverse characteristics and can intervene at multiple stages of cellular processes, including DNA and RNA synthesis, by inhibiting the activity of both cellular and viral polymerases. In the context of respiratory viruses, NAs such as ALS-8112 have been shown to inhibit the replication of RSV. However, it was discontinued due to safety concerns [148]. For rhinoviruses, candidates like 7DMA, which can inhibit RdRp, have been studied [149]. On the other hand, studies have highlighted 2′-deoxy-2′-fluoro-guanosine as a strong candidate against IAV infections [150]. Another NA under investigation is GS-5734, which has shown efficacy against SARS-CoV, SARS-CoV-2, MERS-CoV, and RSV [151,152]. Approximately 25 NAs targeting viral infections have been approved by the FDA [153,154] (Figure 2).

### 10.2. Natural Compounds

Natural compounds have been used for centuries to address various non-infectious diseases. In recent years, their application as potential antivirals against respiratory viruses has gained significant interest [155]. For example, flavonoids isolated from the leaves of *Torreya nucifera* have demonstrated a substantial reduction in SARS-CoV infection in in vitro assays. Similarly, quercetin, a flavonoid found in onions and apples, has shown inhibitory activity against SARS-CoV and SARS-CoV-2 by targeting the main protease, a crucial enzyme for viral replication. Another promising natural compound is glycyrrhizin, derived from licorice (*Glycyrrhiza glabra*), which has exhibited antiviral properties by inhibiting viral replication and modulating the host immune response against SARS-CoV [144,156].

Additionally, saikosaponin D, derived from plants like *Bupleurum* spp., has demonstrated efficacy in blocking viral entry by interfering with the attachment of SARS-CoV to host cells. Essential oils, such as eugenol (from clove) and eucalyptus oil, have also been investigated for their ability to inhibit SARS-CoV-2 replication and entry into pulmonary cells. These natural extracts hold the potential for immune support and are adjuncts in antiviral therapy [157,158].

Beyond SARS-CoV-2, other natural compounds show promise against respiratory viruses like influenza and RSV. For instance, tapsigargin, a plant-derived antiviral, has exhibited broad-spectrum efficacy against SARS-CoV-2, RSV, and influenza A virus by triggering innate immune responses and preventing viral replication at low doses. Moreover, in preclinical studies, sulfated polysaccharides, such as dextran sulfate, have demonstrated potent antiviral effects against SARS-CoV-2 and may hold potential for broader applications against other respiratory pathogens [159].

### 10.3. Peptides

In design, peptides have been classified as fusion or membrane inhibitors [160]. The primary function of anti-fusion peptides is to block the ability of viral proteins to fuse with host cells, thereby impacting the conformational changes of the viral protein [161]. Recently, a fusion peptide was evaluated against SARS-CoV-2 and its spike protein. In vitro assays significantly reduced infection, and ex vivo models indicated that it effectively blocked viral propagation [162]. In addition, the cathelicidin-derived peptide PMAP-36R has shown potent inhibition of SARS-CoV-2 in vitro assays [163].

Advancements have been made in developing peptides to combat IAV infections, such as P7, which blocks viral entry, and P9, which inhibits changes in endosomal pH [164,165]. Another virus studied for applying these peptide advancements is RSV, where a peptide derived from the HR2 domain of the F protein has been explored [166]. Recent modifications of the P9 peptide, in combination with β-defensin-4, have resulted in the creation of P9R. This peptide retains the same properties as P9 but demonstrates a broader antiviral spectrum against SARS-CoV-2, MERS-CoV, SARS-CoV, A(H1N1), A(H7N9), and non-enveloped rhinoviruses [167] (Figure 2).

On the other hand, membrane inhibitors destabilize the viral membrane through the action of specifically designed peptides, altering the viral particle. This property makes them more effective inhibitors [160]. These peptides create pores in the membrane and exhibit this capability only in membranes that do not exceed 160 nm in thickness [128,129,130].

Conducting direct research on these antiviral alternatives is crucial due to the potential for viruses to develop resistance through their evasion strategies and the mutation of viral proteins.

## 11. Conclusions

The development of antivirals against respiratory viruses remains an urgent and challenging priority in modern medicine, especially in the face of the emergence of new variants and increasing viral resistance. Although current antivirals have proven to be valuable tools in mitigating the viral load and severity of respiratory diseases, significant challenges limit their efficacy, such as the ability of viruses to mutate rapidly and the need to improve the accessibility and safety of these treatments for vulnerable populations.

Future research should focus on combining these innovative strategies with a multidisciplinary approach that optimizes the efficacy and specificity of antiviral treatments. Integrating bioinformatics tools, drug repurposing, and exploring natural and peptide compounds present promising avenues for developing effective and sustainable therapies. Thus, science and technology continue to be essential pillars in the fight against respiratory viral infections that threaten global public health.

## Figures and Tables

**Figure 1 pathogens-14-00020-f001:**
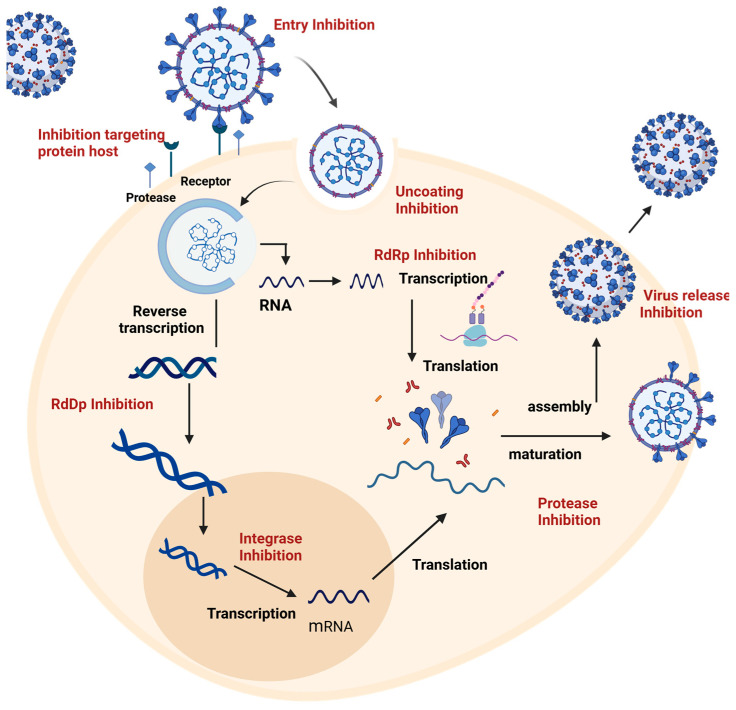
**Viral replication cycle and sites of action of antivirals in respiratory viruses.** The critical phases of the respiratory virus replication cycle are illustrated, including viral binding to cellular receptors, entry into the host by fusion or endocytosis, replication and transcription of viral genetic material, assembly of new virions, and their release from the infected cell. Each phase represents a potential therapeutic target for antivirals. Entry inhibitors block viral binding and fusion with the cell; replication inhibitors act on the viral polymerase, limiting the synthesis of viral RNA or DNA; and assembly and release inhibitors prevent the formation and release of new virions, thus preventing the spread of infection. Antivirals targeting host cellular components are also illustrated as a strategy to interfere with the viral cycle and reduce the possibility of viral resistance.

**Figure 2 pathogens-14-00020-f002:**
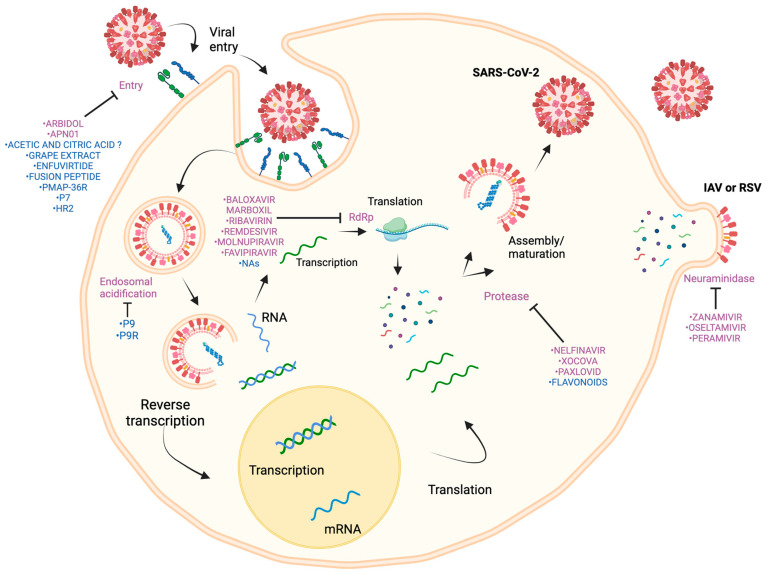
**Mechanism of antiviral action.** The replication cycle of the main viruses SARS-CoV-2, IAV, and RSV is depicted in a simplified form. Principal blocking actions are shown viral entry, endosome acidification, RdRp activity, protease function, and viral release (via neuraminidase). FDA-approved antivirals are shown in purple, while potential future alternatives currently in vitro or clinical testing for these central viral infections are shown in blue. **PMAP-36R:** cathelicidin-derived peptide PMAP-36R. **P7 and P9:** Anti-hemagglutinin antibody-derived peptide 7 or 9. **HR2:** Synthetic peptides derived from the heptad repeat (HR) of fusion (F) proteins. **P9R:** combination of P9 and β-defensin-4.

**Table 1 pathogens-14-00020-t001:** Human respiratory viruses and their characteristics.

Baltimore Group	Family	Virus	Types	Symptomatology	Most Affected Age Group	Complications	Incidence Peaks
I (dsDNA)	Adenoviridae	Adenovirus (HAdV)	A–G	Symptoms of a cold—runny nose, watery diarrhea that comes on suddenly, sore throat, fever, severe cough, vomiting, swollen lymph nodes, headache, and conjunctivitis.	Children < 4 years of age and immunocompromised patients	Pneumonia, bronchitis, encephalitis, meningitis, myocarditis and cardiomyopathies, pulmonary dysplasia, intestinal intussusception, pancreatitis, hemorrhagic cystitis, hepatitis, and nephritis	Winter or early spring [26]
II (ssDNA)	Parvoviridae	Human bocavirus	1–4	Fever, cough, dyspnea, rhinorrhea, diarrhea, wheezing, and pulmonary rales	Children < 5 years	Pneumonia, bronchiolitis, pulmonary atelectasis, and gastrointestinal disorders	Winter or early spring [27]
IV (ssRNA +)	Picornaviridae	Human rhinovirus (HRV)	A–C	Rhinorrhea, nasal congestion, sore throat, cough, headache, subjective fever, and general malaise	Children < 4 years	Acute otitis media, rhinosinusitis, bronchiolitis, pneumonia, cystic or pulmonary fibrosis, and chronic obstructive disease	Spring, summer, and autumn [28]
		Enterovirus D68	-	Fever and cough, wheezing, hypoxia, dysphagia, and dyspnea	Children < 5 years of age and immunocompromised patients	Interstitial pneumonia with diffuse alveolar infiltration and patches in the lungs, asthma exacerbation, and acute flaccid myelitis	Year-round, with a summer resurgence [29]
	Coronaviridae	229E, OC43, NL63, HKU1, SARS-CoV, MERS-CoV, SARS-CoV-2	-	Fever, cough, shortness of breath, anosmia, muscle pain, confusion, headache, sore throat, rhinorrhea, chest pain, diarrhea, nausea, and vomiting	All ages	Viral pneumonia, pulmonary thromboembolism, disseminated intravascular coagulation, encephalitis, vasculitis, ischemic stroke, polyneuropathy, heart failure, arrhythmias, cardiomyopathies, and cardiomegaly	Year-round, higher incidence in winter [8,30,31]
V (ssRNA −)	Pneumovirus	Respiratory syncytial virus (RSV)	A–B	Febrile or afebrile, cough, obstructive dyspnea with polypnea, indrawing, chest overdistention (clinical and/or radiological), and wheezing and/or subcrepitant rales of predominantly expiratory nature	All ages are highly prevalent in children < 5 years and adults > 65 years.	Rhinitis, otitis, laryngitis, bronchitis, bronchiolitis, and pneumonia	Winter in tropical locations, and year-round in warm regions [32,33]
		Human metapneumovirus (hMPV)	A and B, each with subclasses I–II	Common symptoms of upper respiratory tract infection include cough, rhinorrhea, congestion, and sore throat; symptoms of lower respiratory tract infection include wheezing, fever, cough, dyspnea, and hypoxia	Highly prevalent in children < 10 years and adults > 65 years.	Bronchiolitis and pneumonia, asthma exacerbation, or acute respiratory distress syndrome	In the northern hemisphere, it occurs in late winter and early spring [34]
	Paramyxoviridae	Human parainfluenza (PIV)	1–4	Fever, discharge of mucus, cough, and sore throat	Highly prevalent in children < 5 years and adults > 65 years.	Croup, bronchiolitis, bronchitis, and pneumonia	Winter in tropical locations, and year-round in warm regions [35]
	Orthomyxoviridae	Influenzavirus (IAV)	IAV A HA (H1–H18) and 11 NA (N1–N11) and two recent lineages, ‘Victoria’ and ‘Yamagata’, from IAV B	Fever, myalgia, headache, and fatigue	All ages are highly prevalent in children < 5 years and adults > 65 years	Bronchiolitis in children can aggravate asthma, COPD, and congestive heart failure in adults	Peaks of higher incidence during the winter months, depending on the geographic region [6,36]

## Data Availability

Not required.

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
