# Peer review of "Advances and Challenges in Antiviral Development for Respiratory Viruses"

_pathogens, 2024, doi:10.3390/pathogens14010020_

Round 1
Reviewer 1 Report
Comments and Suggestions for Authors
Within the attached review.

Comments are included in the attached review.
Author Response
The authors present a comprehensive review of current and former antivirals against respiratory viruses and offer technology platforms that can be used to uncover novel biological pathways (viral and host). The manuscript is informative and will be appropriate for the Pathogens journal with additional information requested. Specific comments are listed below to facilitate revision. The abstract suggested host factors would be a focus within the review; however, there was little attention outside of brief mention within the COVID-19 section. If not focused on more, please revise the abstract to deemphasize host factor antivirals. Where noted below, there are areas of repetition to be addressed and minor editorial changes to incorporate.
R: Thanks for your comments. We have substantially improved the writing to cover all aspects of the abstract. These aspects are underlined throughout the text.
Specific editorial comments:
- Line 45: add “virus” after respiratory syncytial
R: Thanks for your comments. We have added “virus” in line 45.
- Enlarge Figure 1 for publication
R: Thanks for your comments. We have enlarged Figure 1.
- Lines 74-78, 89-97, and areas within 109-120 are repetitious with regard to antiviral targets being related to protein transcription, translation, assembly, and virus release. Please consolidate.
R: Thanks for your comments. We have consolidated the paragraphs.
- Refer to Table 1 in the text and provide a caption/title for the table. Line 136 is an instruction to authors.
R: Thanks for your comments. We have referred and named Table 1.
- In table 1: RSV and hMPV are pneumoviruses rather than paramyxoviruses based on latest taxonomy.
Please revise and locate RSV and hMPV adjacent to each other.
R: Thanks for your comments. We have made the changes in Table 1.
- Table 1: provide genome description (segmented -ssRNA; Baltimore) group for orthomyxoviridae.
R: Thanks for your comments. The orthomyxoviridae is already in the genome description and Baltimore classification (V).
- Line 167: please verify if Zanamivir can be provided to ≥5 years old (according to CDC website) rather than 7 years old.
R: Thanks for your comments. We have supplemented the information.
- Line 173: Zanamivir is primarily delivered by intranasal route, but can also be administered by IV.
R: Thanks for your comments. We have supplemented the information.
- Line 211: RSV clinical trials have shown a lack of antiviral efficacy in real-world settings which is why there are no other antivirals approved. The sentence implies viral resistance to the antiviral is the cause, which could be phrased more appropriately to lack of efficacy.
R: Thanks for your comments. We have rephrased the sentence.
- Lines 214-216: the clinical trial status will likely change before publication as this is an active field. I recommend removing the detail and noting there are multiple antiviral compounds at various stages of clinical development.
R: Thanks for your comments. We have changed the sentence.
- Line 231: Replace “The invasion mechanism” with “The entry” to avoid colloquialisms.
R: Thanks for your comments. We have changed the sentence.
- Lines 248-249: The FDA and EMA approvals of remdesivir as a COVID-19 treatment supersede the emergency use authorization. Please remove. Also, remdesivir is also a prodrug, which was emphasized for EIDD-2801.
R: Thanks for your comments. We have supplemented the information.
- Line 253 and beyond: Please elaborate on resistance in the clinic to all antivirals.
R: Thanks for your comments. We have supplemented the information in section 7. Clinical Resistance to Antivirals: Mechanisms and Solutions
- Section 5.1.4: The mechanisms of action for RBV are not selective for SARS-CoV-2 and should be summarized once for the manuscript. Further, the viral mutagenesis MOA for molnupiravir is the same as that of ribavirin and should be discussed in Section 5.1.2.
R: Thanks for your comments. We have consolidated the repeated information between molnupiravir and ribavirin and their use in COVID-19.
- Section 5.2.1: The following request is essential to proper coverage of SARS-CoV-2 antivirals. This section should focus on Paxlovid (nirmatrelvir + ritonavir) and Xocova rather than lopinavir, ritonavir, atazanavir, and darunavir as the later compounds are not approved and have shown limited or no efficacy against SARS-CoV-2 in patients. Further, the cellular testing of lopinavir and ritonavir show no potency of these agents against SARS-CoV-2, which suggests any COVID-19 clinical benefit they may have is unrelated to an antiviral effect. It is recommended to reserve discussion of atazanavir and darunavir until clinical efficacy has verified their antiviral effects. Discussion of Viracept should be placed outside of the protease inhibitor section since it is not a known protease inhibitor.
R: Thank you for your comments. We have condensed the sections on lopinavir, ritonavir, atazanavir, and darunavir and added their questionable efficacy to the discussion. In addition, the sections on Paxlovid and Xocova were added.
In the case of Nelfinavir, the following was added: “Nelfinavir has been found to inhibit the activity of the main protease of SARS-CoV-2. However, it is also a potent inhibitor of cell fusion caused by the S glycoprotein [95–97]. “
- Line 335-340: Unclear sentences. Are inhibitors of virus entry intended to block the primary infection?
Also, a stable infection is different than an infectious event. Stable infections are typically not lytic, which SARS-CoV-2 is. Also unclear is use of “union” – is the author intending to describe binding?
R: Thank you for your comments. We have rewritten section 5.3, Inhibitors of virus entry, to better understand it.
- Section 5.3.1: The clinical use of hydroxychloroquine was revoked and not recommended by the WHO following an in-depth analysis. Please provide references to support the mechanism that hydroxychloroquine inhibits ACE2 glycosylation. References 64 and 70 are not the primary literature.
R: Thank you for your comments. The negative findings and clinical reversal of hydroxychloroquine have been added to the text. In addition, the requested relevant citations supporting the drug's mechanism on ACE2 have been added.
- Lines 457-467: One sentence full of multiple concepts that are more of an introduction to the manuscript.
Can this be condensed to the parts that are pertinent to innovations section? Lines 468-473 are also a single quite complex sentence whose message can be focused more for this review.
R: Thank you for your comments. General ideas about respiratory virus challenges have been reduced, keeping only those relevant to the innovation context. The description of recent technologies and their key objectives has been prioritized.
- The reminder of Section 7 is single-sentence paragraphs with many good points; however, please provide additional depth of information within each point made. These seem like bullet points that were combined into once sentence rather than a review of information for a reader. Please revise.
R: Thank you for your comments. The discussion of technologies such as intranasal delivery, nanobodies, stable formulations, and CRISPR/Cas has been expanded and organized, highlighting how these innovations in drug delivery, inhalable vaccines, and gene editing address the specific challenges of respiratory viruses.
- Lines 508-516: this area does not seem tied to innovations and technical advancements. Please relate how the technologies are being applied to resistance or remove. Perhaps this belongs in Section 8. The host-factor antivirals would also combat resistance and are not presented in this section. There is some information in the COVID-19 section (arbidol); however, that is insufficient.
R: Thank you for your comments. We have added more information
- Section 8.1: please add more information on and focus on nucleoside analogs for respiratory viruses.
R: Thank you for your comments. We have added more information in section 9.1
- Section 8.2: please focus on respiratory viruses. The section mostly highlights natural substances as antivirals hepatitis A and E viruses, which are viruses of the liver. Perhaps these are the authors examples of natural substance antivirals, but the relationship to the future exploration of such agents for respiratory viruses is unclear.
R: Thank you for your comments. Examples of compounds specifically investigated for SARS-CoV-2 (quercetin, glycyrrhizin, saikosaponin D), influenza, and RSV (thapsigargin, dextran sulfate) were included.
- Section 8.3: there are many examples of respiratory virus peptide inhibitors; however, the author has only provided one example and extended information to other viruses beyond respiratory. Please focus on respiratory virus peptide inhibitors.
R: Thank you for your comments. We have added more information in section 9.3
Reviewer 2 Report
Comments and Suggestions for Authors
The paper "The Development of Antivirals for Respiratory Viruses: Innovations and Challenges" can be overall divided in two main parts.
1) the authors review antiviral drugs (properties and mechanisms) with focus on influenza, RSV and Sars-Cov-2. From the table that the own authors provide, it is clear that there are many families/types of viruses related to respiratory symptoms. But it is not clear why the authors decided to talk about only 3 of them. A clear rational for this is expected. Well defining the title, the abstract and the scope that it is being presented in the review is essential, so that the reader is prepared for what's coming. Specially in terms of knowing how is the text organized (structure of topics).
2) they attempt to provide an overview of novel approaches, innovations and future for antivirals development. It is hard to differentiate the novel approaches and innovations in the way they are presented, raising the question: could they actually be merged in one single topic? I detected some difficulty in reading these sessions due to redundant (not too informative) and poor english writing. Also, it feels like there are few, limited and not too relevant references used to address the subjects.
It seems that the review's objective is very relevant and could bring some positive impacts into the field. In this regard, I congratulate and appreciate the author's efforts. However, I found the following recent review paper titles: Current state and challenges in respiratory syncytial virus drug discovery and development (2023), A review: Mechanism of action of antiviral drugs (2021), Broad-spectrum coronavirus antiviral drug discovery (2019). To mention a few. So, this is a topic that seem to be well addressed in literature. I'm not saying the attempt of the authors is useless. But they should put more efforts to address their text into a innovative and creative way. They could bring novelty in the structure/organization of the review and also inputting some expertise and critical thoughts/opinions while discussing it.
I also strongly suggest the authors reviewing the title. Make it more specific or focused; to what the review is actually bringing to the reader.
With this in mind, I suggest this paper undergo further review considering the specific-major revisions I will list below, line by line:
55: In addition, we explore.... (reformulate this sentence)
57: 'This review highlights the importance of employing molecular modeling and other biotech tools in drug design, with the ultimate goal to optimize....'
60: 'Finally, we present...' I encourage the authors to employ first active voice (instead of passive voice), not throughout the text; but in parts regarding goals/actions taken to write the main subjects in the review.
66: a brief overview (not A brief overview)
Figure 1: increase the text size in the figure
95: Those or These?
from line 89-125: I detect some redundancy. Eg: the term 'fusion and entry' is mentioned 3 times in distant paragraphs. There may be a way of approaching the subjects in a more organized and concise way.
*Moreover, there are some sentences that are too long (eg: line 115-118). I suggest splitting these long sentences in more objective/readable short sentences. *Please review this throughout the manuscript - I will add a * to others detected.
line 139-143: *long sentence
The Table 1 is not cited in the text. Also, it does not have a caption. I suggest briefly mentioning/referring in the text what is the relevance of Table 1 - or what it summarizes. Importantly, it should be organized in a coherent way, in agreement with the sections explored in the review. The reader naturally expects that all rows (virus types) should correspond to a topic discussed below - if it is not the case, this should be clarified and explained the choice of the ones detailed in text. There are a lot of abbreviations that must be addressed in the caption or footnote of the table. Also, what is the 1st column 'Baltimore group'?
152: no need of . after section title
209: no need of . after section title. Also, avoid the use of abbreviations in section title; use Respiratory syncytial virus (RSV) instead.
230: section is number 5 again? review the numeration of sections throughout the paper.
240: either inhibitors or drugs
257: no need of . after section title
372: APNO1 or APN01?
390: I suggest standardizing the term 'drug repositioning' or 'drug repurposing'. Using both in the manuscript might create confusion.
409: physicochemical and pharmacokinetic properties; also, define ADME abbreviation.
421: this sentence describe docking (reference 80 is about docking). It should instead, more broadly, describe how bioinformatics is relevant.
424: molecular docking might be classified within bioinformatics, but that's not usual. It is preferably classified as a method/technique within Molecular Modeling or Computer-Aided Drug Design. Since the next section 6.1.3 is exclusively talking about docking, this one could explore the others (biological networks and genomic data mining). In fact, I suggest also considering : biomedical ontologies and knowledge graphs (e.g. https://doi.org/10.1093/bioinformatics/btaa718).
426: this paragraph could be moved to section 6.1.3 if you decide describing other bioinformatics on the above.
427: why using AutoDock Vina as example? There are so many other docking software available. Mention that there are many others and maybe cite at least one benchmarking paper that compare/assess docking performances.
429: high precision? If you really wanna state this, provide references.
430: openbabel is useful to convert/manipulate chemical structures and pymol is a GUI visualization software. They may help performing a docking exercise but they don't directly relate to a docking task. Therefore, instead of mentioning them in this sentence, it would be much more useful to write about and cite a paper that critically points best practices for performing a docking study.
432: remove 'As for data and protein libraries'
434: mention that these 3D structures are experimentally obtained/solved by different techniques (NMR, Xray, cryoEM)
437: in general, authors should provide more relevant/highlycited references that describe the state of the art of docking methods. In addition, they could provide the official references for each software and links to the online databases (e.g. https://www.rcsb.org/), when applicable.
449: provide more references. there's a lot of papers that conducted screening campaigns using docking to repurpose drugs in the context of Sars-Cov-2 (e.g. https://doi.org/10.4155/fmc-2021-0025) as well as other viruses.
457: this whole paragraph is made of only one sentence!
468-473: *long sentence
477-480: *long sentence and poor written. Missing period at the end of sentence.
491-497: *long sentence and poor written.
517: I suggest the authors to thoroughly review this section "7. Innovations and technological advances in the development of antivirals". The section is poorly written and it is very hard to follow through and read it. Paragraphs made of a single sentence and without clear meaning should be rewritten.
558: remove 'the' before face
577: remove 2 last sentences about credit taxonomy.
Comments on the Quality of English LanguagePlease see comments above.
Author Response
The paper "The Development of Antivirals for Respiratory Viruses: Innovations and Challenges" can be overall divided in two main parts.
1) the authors review antiviral drugs (properties and mechanisms) with focus on influenza, RSV and Sars-Cov-2. From the table that the own authors provide, it is clear that there are many families/types of viruses related to respiratory symptoms. But it is not clear why the authors decided to talk about only 3 of them. A clear rational for this is expected. Well defining the title, the abstract and the scope that it is being presented in the review is essential, so that the reader is prepared for what's coming. Specially in terms of knowing how is the text organized (structure of topics).
R: Thank you for your comments. We added:
“Respiratory viruses, including influenza, respiratory syncytial virus (RSV), SARS-CoV-2, rhinoviruses, and adenoviruses, significantly impact global health. This review focuses on RSV, influenza, and SARS-CoV-2, the only respiratory viruses with approved antiviral therapies for clinical use. In contrast, other respiratory viruses lack specific antiviral treatments and rely solely on supportive care or investigational therapies, underscoring a critical gap in antiviral development.
This review provides a detailed analysis of antiviral drugs, their mechanisms, and clinical applications by narrowing its scope to viruses with approved treatments. It also offers a structured and focused discussion of current strategies for combating respiratory diseases.”
2) they attempt to provide an overview of novel approaches, innovations and future for antivirals development. It is hard to differentiate the novel approaches and innovations in the way they are presented, raising the question: could they actually be merged in one single topic? I detected some difficulty in reading these sessions due to redundant (not too informative) and poor english writing. Also, it feels like there are few, limited and not too relevant references used to address the subjects.
It seems that the review's objective is very relevant and could bring some positive impacts into the field. In this regard, I congratulate and appreciate the author's efforts. However, I found the following recent review paper titles: Current state and challenges in respiratory syncytial virus drug discovery and development (2023), A review: Mechanism of action of antiviral drugs (2021), Broad-spectrum coronavirus antiviral drug discovery (2019). To mention a few. So, this is a topic that seem to be well addressed in literature. I'm not saying the attempt of the authors is useless. But they should put more efforts to address their text into a innovative and creative way. They could bring novelty in the structure/organization of the review and also inputting some expertise and critical thoughts/opinions while discussing it.
I also strongly suggest the authors reviewing the title. Make it more specific or focused; to what the review is actually bringing to the reader.
R: Thank you for your comments. We appreciate your comments and observations, which have been considered in the new version of the manuscript. You can see these changes highlighted in yellow throughout the text. We consider your comment relevant to the title, so we changed it to Advances and Challenges in Antiviral Development for Respiratory Viruses.
With this in mind, I suggest this paper undergo further review considering the specific-major revisions I will list below, line by line:
55: In addition, we explore.... (reformulate this sentence)
R: Thanks for your comments. We have reformulated the sentence.
57: 'This review highlights the importance of employing molecular modeling and other biotech tools in drug design, with the ultimate goal to optimize....'
R: Thanks for your comments. We have rephrased the sentence.
60: 'Finally, we present...' I encourage the authors to employ first active voice (instead of passive voice), not throughout the text; but in parts regarding goals/actions taken to write the main subjects in the review.
R: Thanks for your comments. We have rephrased the paragraph.
66: a brief overview (not A brief overview)
R: Thanks for your comments. We have changed the sentence
Figure 1: increase the text size in the figure
R: Thanks for your comments. We have enlarged Figure 1.
95: Those or These?
R: Thanks for your comments. We have changed the sentence
from line 89-125: I detect some redundancy. Eg: the term 'fusion and entry' is mentioned 3 times in distant paragraphs. There may be a way of approaching the subjects in a more organized and concise way.
*Moreover, there are some sentences that are too long (eg: line 115-118). I suggest splitting these long sentences in more objective/readable short sentences. *Please review this throughout the manuscript - I will add a * to others detected.
R: Thanks for your comments. We have corrected the text and reviewed the entire document.
line 139-143: *long sentence
The Table 1 is not cited in the text. Also, it does not have a caption. I suggest briefly mentioning/referring in the text what is the relevance of Table 1 - or what it summarizes. Importantly, it should be organized in a coherent way, in agreement with the sections explored in the review. The reader naturally expects that all rows (virus types) should correspond to a topic discussed below - if it is not the case, this should be clarified and explained the choice of the ones detailed in text. There are a lot of abbreviations that must be addressed in the caption or footnote of the table. Also, what is the 1st column 'Baltimore group'?
R: Thanks for your comments. We have made the correction.
152: no need of . after section title
R=Thank you for your comment, we have removed the period.
209: no need of . after section title. Also, avoid the use of abbreviations in section title; use Respiratory syncytial virus (RSV) instead.
R=Thank you for your comment, we have removed the period and corrected the abbreviation.
230: section is number 5 again? review the numeration of sections throughout the paper.
R=Thank you for your comment, we have reviewed the numbering.
240: either inhibitors or drugs
R=Thank you for your comment, we have corrected the sentence.
257: no need of . after section title
R=Thank you for your comment, we have removed the period.
372: APNO1 or APN01?
R= Thank you for your comment, according to the current scientific context, the correct term is APN01.
390: I suggest standardizing the term 'drug repositioning' or 'drug repurposing'. Using both in the manuscript might create confusion.
R=Thank you for your comment, we have corrected the term.
409: physicochemical and pharmacokinetic properties; also, define ADME abbreviation.
R=Thank you for your comment, we have corrected the sentence.
421: this sentence describe docking (reference 80 is about docking). It should instead, more broadly, describe how bioinformatics is relevant.
R=Thank you for your comment, we have corrected the paragraph.
424: molecular docking might be classified within bioinformatics, but that's not usual. It is preferably classified as a method/technique within Molecular Modeling or Computer-Aided Drug Design. Since the next section 6.1.3 is exclusively talking about docking, this one could explore the others (biological networks and genomic data mining). In fact, I suggest also considering : biomedical ontologies and knowledge graphs (e.g. https://doi.org/10.1093/bioinformatics/btaa718).
R=Thank you for your comment, we have made the correction.
426: this paragraph could be moved to section 6.1.3 if you decide describing other bioinformatics on the above.
R=Thank you for your comment, we have made the correction.
427: why using AutoDock Vina as example? There are so many other docking software available. Mention that there are many others and maybe cite at least one benchmarking paper that compare/assess docking performances.
R=Thank you for your comment, we have supplemented the information.
429: high precision? If you really wanna state this, provide references.
R=Thank you for your comment, we have supplemented the information and added a quote.
430: openbabel is useful to convert/manipulate chemical structures and pymol is a GUI visualization software. They may help performing a docking exercise but they don't directly relate to a docking task. Therefore, instead of mentioning them in this sentence, it would be much more useful to write about and cite a paper that critically points best practices for performing a docking study.
R=Thank you for your comment, we have corrected.
432: remove 'As for data and protein libraries'
R=Thank you for your comment, we have removed the sentence.
434: mention that these 3D structures are experimentally obtained/solved by different techniques (NMR, Xray, cryoEM)
R: Thank you for your comment; we have corrected.
437: in general, authors should provide more relevant/highlycited references that describe the state of the art of docking methods. In addition, they could provide the official references for each software and links to the online databases (e.g. https://www.rcsb.org/), when applicable.
R: Thank you for your comment; we have corrected.
449: provide more references. there's a lot of papers that conducted screening campaigns using docking to repurpose drugs in the context of Sars-Cov-2 (e.g. https://doi.org/10.4155/fmc-2021-0025) as well as other viruses.
Thank you for your comment; we have supplemented the information and added a quote.
457: this whole paragraph is made of only one sentence!
R=Thank you for your comment, we have corrected.
468-473: *long sentence
R=Thank you for your comment, we have corrected.
477-480: *long sentence and poor written. Missing period at the end of sentence.
R=Thank you for your comment, we have corrected.
491-497: *long sentence and poor written.
R=Thank you for your comment, we have corrected.
517: I suggest the authors to thoroughly review this section "7. Innovations and technological advances in the development of antivirals". The section is poorly written and it is very hard to follow through and read it. Paragraphs made of a single sentence and without clear meaning should be rewritten.
R=Thank you for your comment. We have rewritten section 7, now named section 9
558: remove 'the' before face
R: Thanks for your comments. We have changed the sentence
577: remove 2 last sentences about credit taxonomy.
R: Thanks for your comments. We have changed the sentence
Reviewer 3 Report
Comments and Suggestions for Authors
The manuscript reviews the most relevant information about the insights and challenges on the development of antivirals for respiratory viruses. The manuscript also gives examples on different clinical therapies used in this field, which in some cases combine traditional and biotechnological approaches.
Typos:
Lines 91-92: toward -> towards
Line 136: Heading of Table 1 is incorrect
Table 1: at least the division lines between rows should be visible to make its analysis more bearable
General comment:
The manuscript contains just one table and one figure. Usually, Review articles provide a greater number of Tables and Figures, in order to show the rich information at a glance. Tables and Figures are a key component of review articles, and, in this case, this is a weakness. I don't know if the authors could strengthen their manuscript with at least one more table or figure. I would really appreciate if they can improve this aspect.
Author Response
The manuscript reviews the most relevant information about the insights and challenges on the development of antivirals for respiratory viruses. The manuscript also gives examples on different clinical therapies used in this field, which in some cases combine traditional and biotechnological approaches.
Typos:
Lines 91-92: toward -> towards
Line 136: Heading of Table 1 is incorrect
Table 1: at least the division lines between rows should be visible to make its analysis more bearable
R: Thanks for your comments. We have changed what was requested
General comment:
The manuscript contains just one table and one figure. Usually, Review articles provide a greater number of Tables and Figures, in order to show the rich information at a glance. Tables and Figures are a key component of review articles, and, in this case, this is a weakness. I don't know if the authors could strengthen their manuscript with at least one more table or figure. I would really appreciate if they can improve this aspect.
R: Thanks for your comments. We added figure 2.
Round 2
Reviewer 1 Report
Comments and Suggestions for Authors
This reviewer appreciates the additional information included in the revised manuscript, which have enhanced the value of the review to the audience. Specific comments and recommendations are provided to address during the editorial process.
Line 60: Retain present tense in introduction. “We discussed” should be “We discuss”
Line 106: Hepatitis is a medical condition rather than a virus. The type of hepatitis virus needs to be specified.
Lines 106-108: The focus of the manuscript is respiratory viruses. Hepatitis viruses (specify), dengue, and HIV are not respiratory viruses and are out of scope with the article. There are examples of entry inhibitors for flu (i.e. DAS181 and neumifil) as well as other respiratory viruses that could be used as examples instead.
Figure 2 legend: please define acronyms used in the figure (for example PMAP-36R, P7, HR2, P9, P9R)
Line 173: The influenza polymerase complex is composed of the PB1 RdRp, PB2 cap binding, and PA endonuclease. Defining PA as the polymerase is incorrect and should be modified for accuracy.
Line 325: Paxlovid is composed of two different drugs, nirmatrelvir and ritonavir. Ritonavir was used previously in HCV and HIV regimens. Paxlovid as not used for HCV or HIV. Please modify for accuracy.
Line 327: please change (3C-like) to (3C-like; 3CL) as this will help with line 348 abbreviation.
Line 331: Remove “on the other hand” as there is no counterpoint.
Lines 391-396 seem redundant to the earlier review of ritonavir as part of a Paxlovid combo. Recommend moving line 391 and the sentence starting on line 394 to the prior paragraph and removing the reminder of the information.
Line 469: remove “and lack of proofreading mechanisms” as SARS-CoV-2 has an exoribonuclease proofreading function in Nsp14. The sentence could also be altered to specify which viruses have no proofreading mechanism and that SARS-CoV-2 does.
Line 496: different font for “time”
Lines 541-555: authors note that the docking studies highlight their importance and best practices, but there is no information on why, what, or how that the reader can take away. A description of why or what these tell us can educate the reader.
Line 544: change “provides” to “provide”
Line 644: ALS-8112 was discontinued due to safety concerns.
Line 647: Please check the name of the compound. 20’ is not possible for a modification of a sugar that has only 1’-5’.
Line 648: Typo: GS-5737 is actually GS-5734, which is remdesivir and should be noted as such. This nucleoside prodrug is also potent against other respiratory viruses, including RSV and others. Please review literature and include appropriate respiratory viruses.
Line 649 is not necessary for a respiratory virus focus as HSV, HCV, and HIV are not respiratory viruses. The same comment applies to lines 679-681.
Lines 696-698: Out of scope for respiratory viruses and should be removed.
Author Response
This reviewer appreciates the additional information included in the revised manuscript, which have enhanced the value of the review to the audience. Specific comments and recommendations are provided to address during the editorial process.
Line 60: Retain present tense in introduction. “We discussed” should be “We discuss”
R: Thank you for your comments. We have made the suggested changes, which are marked in green.
Line 106: Hepatitis is a medical condition rather than a virus. The type of hepatitis virus needs to be specified.
R: Thank you for your comments. We have made the suggested changes.
Figure 2 legend: please define acronyms used in the figure (for example PMAP-36R, P7, HR2, P9, P9R)
R: Thank you for your comments. We have made the suggested changes.
Line 173: The influenza polymerase complex is composed of the PB1 RdRp, PB2 cap binding, and PA endonuclease. Defining PA as the polymerase is incorrect and should be modified for accuracy.
R: Thank you for your comments. We have made the suggested changes.
Line 325: Paxlovid is composed of two different drugs, nirmatrelvir and ritonavir. Ritonavir was used previously in HCV and HIV regimens. Paxlovid as not used for HCV or HIV. Please modify for accuracy.
R: Thank you for your comments. We have made the suggested changes.
Line 327: please change (3C-like) to (3C-like; 3CL) as this will help with line 348 abbreviation.
R: Thank you for your comments. We have made the suggested changes.
Line 331: Remove “on the other hand” as there is no counterpoint.
R: Thank you for your comments. We have made the suggested changes.
Lines 391-396 seem redundant to the earlier review of ritonavir as part of a Paxlovid combo. Recommend moving line 391 and the sentence starting on line 394 to the prior paragraph and removing the reminder of the information.
R: Thank you for your comments. We have made the suggested changes.
Line 469: remove “and lack of proofreading mechanisms” as SARS-CoV-2 has an exoribonuclease proofreading function in Nsp14. The sentence could also be altered to specify which viruses have no proofreading mechanism and that SARS-CoV-2 does.
R: Thank you for your comments. We have made the suggested changes.
Line 496: different font for “time”
R: Thank you for your comments. We have made the suggested changes.
Lines 541-555: authors note that the docking studies highlight their importance and best practices, but there is no information on why, what, or how that the reader can take away. A description of why or what these tell us can educate the reader.
R: Thank you for your comments. We have added:
“For more information, we recently published a chapter on drug repurposing against emerging viruses, describing the importance of using molecular docking in antiviral discovery [130].”
Line 544: change “provides” to “provide”
R: Thank you for your comments. We have made the suggested changes.
Line 644: ALS-8112 was discontinued due to safety concerns.
R: Thank you for your comments. We have made the suggested changes.
Line 647: Please check the name of the compound. 20’ is not possible for a modification of a sugar that has only 1’-5’.
R: Thank you for your comments. We have made the suggested changes.
Line 648: Typo: GS-5737 is actually GS-5734, which is remdesivir and should be noted as such. This nucleoside prodrug is also potent against other respiratory viruses, including RSV and others. Please review literature and include appropriate respiratory viruses.
R: Thank you for your comments. We have made the suggested changes.
Line 649 is not necessary for a respiratory virus focus as HSV, HCV, and HIV are not respiratory viruses. The same comment applies to lines 679-681.
R: Thank you for your comments. We have made the suggested changes.
Lines 696-698: Out of scope for respiratory viruses and should be removed.
R: Thank you for your comments. We have made the suggested changes.